# INCENTIVIZING MULTIMODAL REASONING VIA PROGRESSIVE CURRICULUM REINFORCEMENT LEARNING

## ABSTRACT

Reinforcement learning has shown strong potential in improving the reasoning abilities of large language models, and recent studies extend this paradigm to multimodal reasoning. However, the complexity and diversity of multimodal tasks often lead to unstable performance across domains and difficulty levels. To address these challenges, we introduce VL-COGITO, a multimodal reasoning model trained with a multi-stage Progressive Curriculum Reinforcement Learning (PCuRL) framework. PCuRL gradually increases task difficulty, enhancing reasoning robustness in diverse contexts. It features two key innovations: (1) an online difficulty–aware weighting mechanism that dynamically adjusts task difficulty across training stages, and (2) a dynamic length reward that encourages adaptive control of reasoning path length to balance efficiency and accuracy. Experiments demonstrate that VL-COGITO achieves state-of-the-art performance on 8 out of 10 benchmark tasks spanning mathematics, science, logic, and general understanding, while matching comparable results on the remaining 2 tasks, validating the effectiveness of our approach.

## 1 INTRODUCTION

Recent advances of Reinforcement Learning (RL) in Large Language Models (LLMs; OpenAI (2024b); Kimi (2025); DeepSeek-AI (2025)) highlight its promise to incentivize LLMs' long-chain reasoning abilities, enabling them to effectively tackle complex tasks like code, mathematics, and sciences. In particular, RL with verifiable rewards, such as GRPO (Shao et al., 2024), has emerged as a pivotal paradigm, which directly employs rule-based rewards and iteratively refines multiple reasoning paths via group-based relative advantage estimations. This approach has notably improved the performance of LLMs, pushing the boundaries of what LLMs can achieve in various domains (Yeo et al., 2025; Chen et al., 2025c).

Following the success, researchers have increasingly explored its application to Multimodal LLMs (MLLMs). Initial efforts primarily focus on adapting these techniques to specific multimodal domains such as mathematics and logic (Meng et al., 2025; Huang et al., 2025; Chen et al., 2025a;b). As MLLMs are not constrained to textual modalities, it has the opportunity to enable reasoning across diverse domains. The spectrum ranges from straightforward chart interpretation (Huang et al., 2024), complex geometry problems (Lu et al., 2024), to intricate scientific analysis (Lu et al., 2022a). With this expanded scope of domains, the heterogeneity among different problem types becomes increasingly apparent, presenting a challenge in effectively learning reasoning skills for tasks of varying complexities and types (Wang et al., 2025e; Bi et al., 2025; Li et al., 2025).

To bridge this gap, we introduce VL-COGITO, a reasoning-oriented MLLM trained on an extensive dataset comprising diverse multimodal task domains. Rather than treating all tasks uniformly, we recognize that varying difficulty levels and domain-specific requirements necessitate a more sophisticated training approach. Therefore, we propose a **P**rogressive **Cu**rriculum **R**einforcement **L**earning framework (**PCuRL**). Central to PCuRL is a novel curriculum learning strategy, which systematically guides the model through progressively complex tasks to build robust reasoning capabilities. Specifically, we introduce an *online difficulty soft weighting* mechanism to dynamically adjust data selection according to difficulty distributions across successive training stages, allowing the model to incrementally transition from mastering simpler questions to effectively handling intricate problems. Another key component is the *dynamic length reward mechanism*. Instead of

indiscriminately extending reasoning length, the dynamic reward explicitly incentivizes the model to adapt its reasoning length based on the demands of individual problems. This strategy ensures the model not only engages in thorough reasoning for complex tasks but also maintains efficiency and succinctness when simpler problems arise, thus optimizing performance across diverse scenarios.

To thoroughly assess the performance of VL-COGITO, we perform extensive experiments on various multimodal reasoning benchmarks spanning mathematical, scientific, and general domains. It is worth noting that VL-COGITO bypasses the cold-start SFT phase and is instead trained directly from the backbone model via PCuRL using GRPO. The results demonstrate that VL-COGITO achieves state-of-the-art or highly competitive performance across all evaluation sets, underscoring the superiority and effectiveness of PCuRL. Furthermore, comprehensive ablation studies are conducted to analyze the contribution of each module, confirming their respective importance. Visualizations of the training process and case studies validate that PCuRL ensures training stability while enhancing both effectiveness and efficiency.

Our contributions are threefold: (1) We propose PCuRL, a novel multi-stage progressive curriculum RL framework, incorporating an online difficulty soft weighting mechanism that progressively exposes models to increasingly challenging tasks, enhancing multimodal reasoning capabilities; and a dynamic length reward mechanism designed to encourage the model to modulate reasoning length based on question-specific complexity, effectively balancing depth and efficiency. (2) VL-COGITO achieves the state-of-the-art or highly competitive performance in diverse multimodal benchmarks, underscoring its effectiveness and versatility. (3) Extensive ablation studies confirm that the progressive curriculum learning mechanism consistently increases reasoning depth, leading to improved performance on complex tasks. The dynamic length reward strategy allows the model to produce concise answers for simple problems while promoting longer, more in-depth reasoning for more difficult ones. Overall, PCuRL delivers substantial and balanced gains in accuracy, training stability, and efficiency across a range of task difficulties.

## 2 RELATED WORK

As RL-based reasoning has proven effective in LLMs, recent research increasingly explores its adaptation to MLLMs (Yao et al., 2024; Xu et al., 2025; Wang et al., 2025c; Peng et al., 2025a; Xia et al., 2025). Several studies directly enhance multimodal reasoning via RLVR-style optimization (Kimi et al., 2025; Wang et al., 2025b; Guo et al., 2025; Team et al., 2025; Huang et al., 2025; Yang et al., 2025; Deng et al., 2025), while others address training stability and effectiveness through data selection (Wang et al., 2025a; Meng et al., 2025), reward design (Shen et al., 2025; Tan et al., 2025), and advanced RL methods (Peng et al., 2025b; Zhang et al., 2025; Liu et al., 2025; Yao et al., 2025; Chen et al., 2025d; Leng et al., 2025). For example, MM-Eureka (Meng et al., 2025) shows that difficulty-based selection is crucial for RL, and ThinkLite-VL (Wang et al., 2025d) leverages Monte Carlo Tree Search to achieve data-efficient training. R1-VL (Zhang et al., 2025) introduces Step-GRPO with step-wise rewards. VL-Rethinker (Wang et al., 2025a) employs selective replay and forced rethinking, and Praxis-VLM (Hu et al., 2025) adapts rewards to different skills across training stages. In this context, we propose a novel curriculum-inspired RL recipe that integrates online difficulty weighting to progressively train on harder tasks and introduces a dynamic length reward that adaptively determines the optimal reasoning length for each prompt.

## 3 PRELIMINARIES

Group Relative Policy Optimization (GRPO; Shao et al. (2024)) is a reinforcement learning algorithm designed to improve the efficiency and effectiveness of training language models. It estimates the advantages of model generations by comparing responses within a group specific to the same input. Given an input $x$, the behavior policy $\pi_{\theta_{\text{old}}}$ first samples a group of $G$ candidate responses $\{y_i\}_{i=1}^G$. At time step $t$, the advantage for the $i$-th response is calculated as:

$$A_{i,t} = \frac{r(x, y_i) - \text{mean}(\{r(x, y_1), \ldots, r(x, y_G)\})}{\text{std}(\{r(x, y_1), \ldots, r(x, y_G)\})}, \tag{1}$$

where the standard reward function is a combination of accuracy and format rewards ($r(x, y_i) = r_{\text{acc}}(x, y_i) + r_{\text{format}}(y_i),$) that evaluates the correctness and format of model outputs. The GRPO also

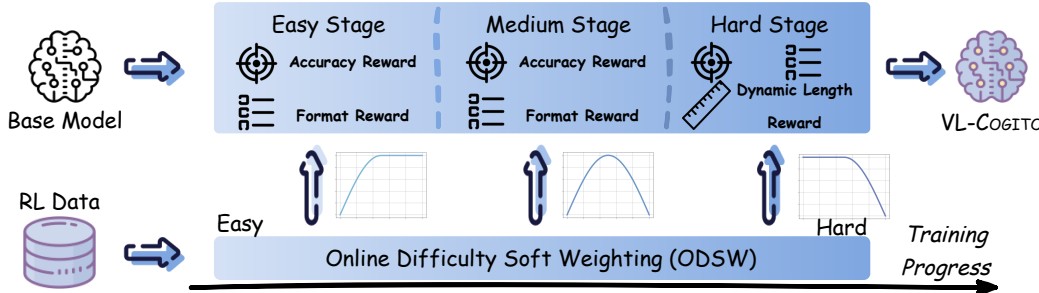

Figure 1: An overview of the PCuRL framework. It consists of two key components: (1) a multi-stage curriculum RL structure that segments training into progressive stages, with online difficulty soft weighting dynamically adjusting prompt advantages based on task complexity; (2) a dynamic length reward mechanism that encourages the model to adapt its reasoning chain length according to task complexity, rather than indiscriminately increasing it.

adopts a clipped surrogate objective like PPO (Schulman et al., 2017):

$$\frac{1}{G}\sum_{i=1}^{G}\frac{1}{|y_i|}\sum_{t=1}^{|y_i|}\min\left[\frac{\pi_\theta(y_{i,t}\mid x,y_{i,<t})}{\pi_{\theta_{\text{old}}}(y_{i,t}\mid x,y_{i,<t})}A_{i,t},\text{clip}\left(\frac{\pi_\theta(y_{i,t}\mid x,y_{i,<t})}{\pi_{\theta_{\text{old}}}(y_{i,t}\mid x,y_{i,<t})},1-\epsilon,1+\epsilon\right)A_{i,t}\right],\quad(2)$$

where $\epsilon$ serves as a hyperparameter that controls the tolerance for policy deviation. The clip function is used to prevent excessively large updates by maintaining the ratio between the current policy and the reference policy within a specified range.

## 4 METHODOLOGY

### 4.1 DATA CURATION

To promote robust performance and strong generalization across a variety of domains, we collect an extensive collection of open-source multimodal datasets, covering 23 datasets distributed across six distinct task categories, *i.e.*, *Mathematical Reasoning*, *Logical Reasoning*, *Counting*, *Science Reasoning*, *Chart Understanding*, and *General Image Understanding*.

We curate the RL training set by selective sampling according to quality and category criteria, aiming to increase task difficulty and coverage to foster deeper reasoning. To this end, we adopt two strategies: (1) **Open-ended Format**: Most samples are reformulated into open-ended QA to avoid reliance on superficial cues from fixed formats (*e.g.*, multiple-choice); (2) **Difficulty Sampling**: We filter out questions that are solvable without genuine reasoning by excluding samples that exceed $50\%$ accuracy over $8$ trials with Qwen2.5-VL-7B (Bai et al., 2025). The details of the dataset collection and resultant distribution of the RL training data are presented in Appendix B.1.

### 4.2 PROGRESSIVE CURRICULUM REINFORCEMENT LEARNING FRAMEWORK

#### 4.2.1 OVERALL FRAMEWORK

Consistent with prior work (Meng et al., 2025; Wang et al., 2025a;d), we leverage the GRPO-based RLVR scheme and propose a **P**rogressive **Cu**rriculum **R**einforcement **L**earning (**PCuRL**) framework. PCuRL progressively directs the model's attention toward increasingly difficult tasks to acquire more sophisticated reasoning skills, while maintaining concise and effective responses on simpler tasks. PCuRL comprises two key components: (1) a multi-stage curriculum RL that utilizes *online difficulty soft weighting*, and (2) a *dynamic length reward mechanism* that adapts the reasoning length according to task complexity, rather than indiscriminately increasing it.

The overall architecture is depicted in Figure 1. The training is organized into three progressive stages: *easy*, *medium*, and *hard*. At each stage, the same dataset is used but is reshuffled to encourage generalization. The difficulty-aware soft weighting mechanism dynamically adjusts training

emphasis throughout the learning process. During the easy stage, it assigns higher weights to simple training examples, while in the hard stage, it prioritizes more challenging examples. The dynamic length reward is applied only in the hard stage, where it incentivizes longer reasoning chains for complex questions. This design achieves two goals: (1) the early stages, without length rewards, encourage rapid adaptation and initial performance gains; and (2) the hard stage, driven by challenging tasks, promotes extended reasoning at the cost of slower convergence compared to earlier stages.

### 4.2.2 ONLINE DIFFICULTY SOFT WEIGHTING

The difficulty soft weighting mechanism prioritizes target prompts by difficulty during RL training. Prior work (Bae et al., 2025; Cui et al., 2025) introduces a hard-filtering approach that discards prompts outside a predefined range. In contrast, our method retains more prompts, assigning each a weight reflecting its relative importance. Inspired by ADORA (Gui & Ren, 2025), we propose **O**nline **D**ifficulty **S**oft **W**eighting (**ODSW**), which adjusts prompt-level advantages according to difficulty-based weights. Prompts with higher weights exert a stronger influence on gradient updates. Specifically, weights are computed from rollout accuracy using a predefined function $F$:

$$\hat{A}_{i,t} = F\Big(\frac{1}{G} \sum_{i=1}^{G} \mathrm{acc}(x, y_i)\Big) \cdot A_{i,t},\tag{3}$$

where $G$ denotes the number of rollout responses per prompt and acc indicates whether a rollout is correct or not (*i.e.*, 1 for correct and 0 for incorrect). The definition of $F$ is based on the theory of learnability (Foster & Foerster, 2025; Rutherford et al., 2024; Tzannetos et al., 2023), which indicates that prompts with rollout accuracy near $0.5$ are optimal for RL training, as they balance challenge and learnability. Guided by this principle, we define $F$ as a *continuous piecewise function* combining sine and constant components with $0.5$ as the threshold. The constant terms anchor the model at the target difficulty, while the sine term ensures smooth transitions across difficulty levels. This design highlights prompts with maximal learnability, directs gradient updates toward a chosen difficulty, and prevents neglect of others. We implement three variants of $F$, tailored to easy, medium, and hard stages. An illustration of ODSW is presented in Appendix B.2. For comparison, we also evaluate a binary weighting strategy alongside our proposed soft-weighting methods. The definitions of these weighting functions are given as follows:

$$F_{\text{Easy}}(\text{Acc}) = \begin{cases} \sin(\pi \cdot \text{Acc}), & \text{Acc} \in [0, 0.5) \\ 1, & \text{Acc} \in [0.5, 1] \end{cases}; F_{\text{Medium}}(\text{Acc}) = \sin(\pi \cdot \text{Acc});$$

$$F_{\text{Hard}}(\text{Acc}) = \begin{cases} 1, & \text{Acc} \in [0, 0.5] \\ \sin(\pi \cdot \text{Acc}), & \text{Acc} \in (0.5, 1] \end{cases}; F_{\text{Binary}}(\text{Acc}) = \begin{cases} 1, & \text{if Acc} \in [T_{\min}, T_{\max}] \\ 0, & \text{otherwise} \end{cases}.$$

$$\tag{4}$$

For simplicity, we denote the average rollout accuracy as $\text{Acc} = \frac{1}{G} \sum_{i=1}^{G} \mathrm{acc}(x, y_i)$. The $T_{\min}$ and $T_{\max}$ in the binary weighting strategy are predefined ranges of average accuracy. By integrating explicit difficulty control with curriculum learning, we facilitate progressive model optimization. Training on simpler tasks first enables the model to establish reliable reasoning patterns and obtain stronger optimization signals to stabilize learning, while transitioning to more complex tasks promotes deeper reasoning and exploration of diverse solution paths, further enhancing performance.

### 4.2.3 DYNAMIC LENGTH REWARD MECHANISM

Extending the reasoning process gives models more room to think and explore diverse strategies for complex problems. A common approach is to apply length rewards, such as Cosine Reward (Yeo et al., 2025). However, these rewards are typically uniform across prompts, overlooking task-specific needs. In multimodal reasoning, optimal reasoning length varies: chart interpretation often requires shorter explanations, while geometry problems may demand longer ones. Applying a single-length target indiscriminately can thus cause unnecessary verbosity in some cases and insufficient reasoning in others, ultimately reducing both efficiency and performance.

To address this issue, we introduce the **Dy**namic **L**ength **R**eward (**DyLR**), which adaptively determines the optimal reasoning length for each prompt. The target length is estimated from rollout samples as the average length of all correct responses for that prompt, guiding the model to produce reasoning paths aligned with this prompt-specific budget. As training progresses and accuracy

improves, these targets evolve dynamically and gradually stabilize. For prompts without correct responses, DyLR encourages the model to extend reasoning up to a predefined maximum. Formally, the dynamic length reward $r_{\text{len}}$ is defined as:

$$r_{\text{len}}(y_i, y) = \begin{cases} \text{CosFn}(L_i, L_{\text{avg}}, r_{\text{len}}^{\min}, r_{\text{len}}^{\max}) & \text{if Acc} > 0 \\ \text{CosFn}(L_i, L_{\text{max}}, r_{\text{len}}^{\min}, r_{\text{len}}^{\max}) & \text{if Acc} = 0 \end{cases}, \tag{5}$$

where $L_i$ denotes the reasoning length of the $i$-th response $y_i$ to a prompt $x$, $L_{\text{avg}}$ is the average length of all correct responses to $x$, and $L_{\text{max}}$ is the preset maximum target length. $r_{\text{len}}^{\min}$ and $r_{\text{len}}^{\max}$ specify the minimum and maximum length rewards, respectively. The function CosFn is defined as:

$$\text{CosFn}(L_i, L_{\text{tgt}}, r_{\text{len}}^{\min}, r_{\text{len}}^{\max}) = r_{\text{len}}^{\min} + \frac{1}{2}(r_{\text{len}}^{\max} - r_{\text{len}}^{\min})\left(1 - \cos\left(\frac{L_i \cdot \pi}{L_{\text{tgt}}}\right)\right), \tag{6}$$

where $L_{\text{tgt}} \in \{L_{\text{avg}}, L_{\text{max}}\}$ is the target reasoning length. Through combining the standard reward of GRPO and dynamic length rewards, the overall reward is defined as:

$$r(x, y_i) = \alpha \cdot r_{\text{acc}}(x, y_i) + \beta \cdot r_{\text{format}}(y_i) + \gamma \cdot r_{\text{len}}(y_i, y), \tag{7}$$

where $\alpha$, $\beta$, and $\gamma$ are hyperparameters to control the contributions of each specific reward term.

To prevent excessive reasoning length from impairing quality, we introduce a hyperparameter $w$ applied to prompts with zero accuracy (*i.e.*, Acc $= 0$). Incorporating this into the dynamic length reward with difficulty soft weighting, Equation 3 becomes:

$$\hat{A}_{i,t} = \begin{cases} F\left(\frac{1}{G}\sum_{i=1}^{G}\text{acc}(x, y_i)\right) \cdot A_{i,t} & \text{if Acc} > 0 \\ w \cdot F\left(\frac{1}{G}\sum_{i=1}^{G}\text{acc}(x, y_i)\right) \cdot A_{i,t} & \text{if Acc} = 0 \end{cases}. \tag{8}$$

In general, DyLR allows the model to adapt its output length dynamically, avoiding fixed targets that risk over- or under-thinking. For simple tasks, it produces concise answers through quick reasoning, while for complex tasks, it extends the reasoning process to ensure accuracy.

## 5 EXPERIMENTS

### 5.1 EXPERIMENTAL SETTINGS

**Benchmark datasets.** To comprehensively assess VL-COGITO, we adopt a diverse suite of multimodal benchmarks: (1) **Math and Logic**: For math, we choose the test set of Geometry@3k (Geo3k; hiyouga (2025a)), MathVision (Wang et al., 2024) test set, the testmini set of MathVista (Lu et al., 2024), and MathVerse (Zhang et al., 2024); For logic, we use LogicVista (Xiao et al., 2024); For chart understanding, we use the test set of ChartQA (Masry et al., 2022). (2) **Science**: The test sets of ScienceQA (SciQA; Lu et al. (2022b), MMMU (Yue et al., 2024), and EMMA (Hao et al., 2025) are selected in our benchmark suite to evaluate the model's capability in the scientific domain. (3) **General Understanding**: we further use a general image understanding benchmark, MMStar (Chen et al., 2024), to measure the model's fundamental visual understanding ability.

**Baselines.** We evaluate VL-COGITO against two categories of comparable MLLMs: (1) **General-purpose models**, including Qwen2.5-VL-7B-Instruct (Bai et al., 2025), InternVL2.5-8B (Chen et al., 2025e), InternVL3-8B (Zhu et al., 2025), and LLaVA-OneVision-7B (LLaVA-OV; Li et al. (2024a)), representing recent SoTA MLLMs; and (2) **Reasoning-oriented models**, such as MM-Eureka-8B (Meng et al., 2025), R1-VL-7B (Zhang et al., 2025), MMR1-7B (Leng et al., 2025), R1-OneVision-7B (Yang et al., 2025), OpenVLThinker-7B (Deng et al., 2025), Vision-R1-7B (Huang et al., 2025), VL-Rethinker (Wang et al., 2025a), and ThinkLite-VL-7B (Wang et al., 2025d).

**Evaluation.** We adopt a unified prompt (ref. Appendix B.3) across all evaluations, requiring models to enclose the final answer within "\box{}". Inference is performed using vLLM (Kwon et al., 2023) for efficient generation. For benchmarks with official evaluation protocols (*e.g.*, MathVision, MMMU), we strictly follow their procedures. For others, mathematical questions are assessed using Math-Verify (Kydlíček, 2025) and MathRuler (hiyouga, 2025b), while non-mathematical ones are evaluated by exact matching. To ensure robustness and fairness, two additional measures are applied: (i) for multiple-choice questions where the model's output does not match any option, the

Table 1: Comparison of VL-COGITO with other MLLMs. All models are reevaluated under identical conditions for fairness. Results affected by benchmark contamination are excluded and marked as "–". The **bold** and underline highlight the best and second-best scores among reasoning models.

| Model | Size | Mathematics | | | | | | Science | | | General | Average |
|---|---|---|---|---|---|---|---|---|---|---|---|---|
| | | Geo3K | MathVerse | MathVista | MathVision | LogicVista | ChartQA | SciQA | MMMU | EMMA | MMStar | |
| General-Purpose Models | | | | | | | | | | | | |
| Qwen2.5-VL | 7B | 61.6 | 50.4 | 69.3 | 28.7 | 44.0 | 82.4 | 85.4 | 50.9 | 24.6 | 62.5 | 56.0 |
| InternVL2.5 | 8B | 60.6 | 40.0 | 61.4 | 19.9 | 37.7 | 73.4 | 90.3 | 43.1 | 19.9 | 62.2 | 50.9 |
| InternVL3 | 8B | 63.3 | 49.4 | 68.5 | 30.0 | 41.3 | 81.3 | 89.3 | 50.8 | 14.5 | 67.5 | 55.6 |
| LLaVA-OV | 7B | 48.5 | 33.6 | 56.4 | 15.9 | 30.6 | 65.0 | 80.5 | 41.6 | 18.3 | 53.5 | 44.4 |
| Reasoning-Oriented Models | | | | | | | | | | | | |
| MM-Eureka | 8B | 67.2 | 52.3 | 73.4 | 29.4 | 47.1 | 82.7 | 86.4 | 52.3 | 27.4 | 64.7 | 58.3 |
| R1-VL | 7B | 57.5 | 41.3 | 61.5 | 23.0 | 36.3 | 76.3 | 86.0 | 38.1 | 24.0 | 55.6 | 50.0 |
| MMR1 | 7B | 65.9 | 52.5 | 73.6 | **32.9** | 46.6 | 82.8 | 86.7 | **53.1** | 28.1 | 66.3 | 58.9 |
| R1-OneVision | 7B | 57.9 | 44.0 | 60.3 | 22.0 | 40.0 | 72.5 | 85.3 | 43.4 | 22.2 | 56.2 | 50.4 |
| OpenVLThinker | 7B | 60.6 | 48.1 | 70.6 | 22.0 | 41.0 | 81.0 | 85.9 | 50.9 | 24.9 | 62.8 | 54.8 |
| VL-Rethinker | 7B | 67.7 | 54.6 | 73.7 | 30.1 | 45.7 | 83.5 | **86.7** | 52.9 | 28.6 | 64.2 | 58.8 |
| Vision-R1 | 7B | 67.0 | 51.9 | 72.1 | - | 44.7 | 82.7 | - | 26.4 | 28.3 | 65.4 | - |
| ThinkLite-VL | 7B | 63.5 | 51.3 | 72.5 | 27.5 | 44.3 | 83.1 | - | 50.9 | 26.4 | 64.6 | - |
| VL-COGITO | 7B | **69.4** | **55.4** | **74.8** | 30.9 | **49.8** | **83.9** | 88.2 | 52.6 | **28.9** | 67.0 | **60.1** |

most semantically similar choice is selected; (ii) for open-ended tasks where exact matching fails or parsers cannot extract answers, GPT-4o (OpenAI, 2024a) is used as an auxiliary judge.

**Implementations.** We adopt Qwen2.5-VL-7B-Instruct as the backbone and train with the AdamW optimizer (learning rate $1 \times 10^{-6}$). For GRPO configuration, we set rollout batch size 512, global batch size 128, maximum sequence length $4,096$, KL coefficient $1 \times 10^{-3}$, and sample 16 responses per prompt at temperature 1.0. The system prompt is provided in Appendix B.3. For PCuRL, **preliminary experiments show that reward and validation accuracy plateau after** 100 **optimization steps in the easy and medium stages due to data simplicity, while the hard stage requires substantially more updates.** Thus, we run 100 steps in the easy/medium stages, selecting the best checkpoint for the next stage, and train for one epoch (200 steps) in the hard stage, where the length reward increases convergence difficulty. Reward hyperparameters are set to $\alpha = 1$, $\beta = 0.5$, $\gamma = 1$, $w = 0.25$. The dynamic length reward is penalized with $r_{\text{len}}^{\min} = -1$ and $r_{\text{len}}^{\max} = 0$. Baselines are directly obtained from HuggingFace and run under the same environment.

## 5.2 RESULTS AND ANALYSES

**Main results.** Table 1 compares VL-COGITO with both general and reasoning MLLMs across 10 multimodal benchmarks. General-purpose models (*e.g.*, InternVL, Qwen2.5-VL) perform well in certain scientific domains but lag behind reasoning-focused MLLMs on math benchmarks that require advanced analytical skills. Compared to its backbone Qwen2.5-VL-Instruct, VL-COGITO delivers consistent gains across mathematics, science, and general tasks, achieving absolute improvements of 7.8%, 5.0%, and 5.8% on Geometry@3K, MathVerse, and LogicVista, and 2.8%, 4.3%, and 4.5% on ScienceQA, EMMA, and MMStar, respectively. These results highlight the robustness and broad applicability of our approach, demonstrating benefits beyond a single domain.

Among reasoning-oriented models, VL-COGITO delivers state-of-the-art or highly competitive results *without requiring a cold-start warm-up*. This highlights the strength of our progressive curriculum learning strategy. It achieves the best performance on 8 of 10 multimodal benchmarks, excelling on demanding mathematics and science tasks while also leading on lighter benchmarks such as ScienceQA and MMStar. Compared to VL-Rethinker, VL-COGITO falls slightly behind only on MMMU, but surpasses it elsewhere. Unlike VL-Rethinker's forced multi-step rethinking during RL training, VL-COGITO relies purely on autonomous exploration. Moreover, while models like R1-VL, R1-OneVision, OpenVLThinker, and Vision-R1 depend on cold-start initialization with carefully curated SFT datasets, VL-COGITO consistently outperforms them. These results validate the effectiveness of our RL pipeline in enhancing reasoning across diverse domains.

**Component-wise performance decomposition of PCuRL.** Table 2 reports the ablation results quantifying the contribution of each PCuRL component. Relative to the vanilla GRPO baseline, VL-COGITO improves performance on reasoning-intensive benchmarks (*i.e.*, MathVision, LogicVista, MMMU, EMMA) while maintaining comparable accuracy on easier tasks (*i.e.*, ChartQA,

Table 2: Component-wise performance decomposition of PCuRL. "+Curriculum" and "+DyLR" represent adding progressive curriculum strategy and dynamic length reward to GRPO, respectively.

| Model | Mathematics | | | | | | Science | | | General | Avg |
|---|---|---|---|---|---|---|---|---|---|---|---|
| | Geo3K | MathVerse | MathVista | MathVision | LogicVista | ChartQA | SciQA | MMMU | EMMA | MMStar | |
| Vanilla GRPO | 66.1 | 52.2 | 71.4 | 30.0 | 44.0 | 83.8 | 87.8 | 51.4 | 28.0 | 66.3 | 58.1 |
| +Curriculum | 67.4 | 51.9 | 74.0 | 30.4 | 47.5 | 83.9 | 87.6 | 52.7 | 28.3 | 65.2 | 58.9 |
| +DyLR | 66.2 | 52.5 | 73.3 | 29.4 | 48.4 | 83.2 | 87.1 | 52.6 | 28.2 | 64.9 | 58.6 |
| VL-COGITO | 69.4 | 55.4 | 74.8 | 30.9 | 49.8 | 83.9 | 88.2 | 52.6 | 28.9 | 67.0 | 60.1 |

Table 3: Ablation study of the online difficulty soft weighting (ODSW). "Binary" denotes the binary weighting, where we assess three difficulty ranges of $[T_{\min}, T_{\max}]$; ODSW "Easy", "Medium", and "Hard" represent only utilizing the three ODSW variants during the RL training, respectively.

| Model | Mathematics | | | | | | Science | | | General | Avg |
|---|---|---|---|---|---|---|---|---|---|---|---|
| | Geo3K | MathVerse | MathVista | MathVision | LogicVista | ChartQA | ScienceQA | MMMU | EMMA | MMStar | |
| Binary $[0.50, 1.00]$ | 59.6 | 49.3 | 72.0 | 27.8 | 41.5 | 82.6 | 87.1 | 49.1 | 25.2 | 64.5 | 55.9 |
| Binary $[0.25, 0.75]$ | 67.2 | 51.9 | 72.3 | 30.4 | 46.7 | 84.7 | 87.6 | 51.0 | 27.0 | 64.4 | 58.3 |
| Binary $[0.00, 0.50]$ | 65.6 | 51.4 | 73.1 | 29.5 | 45.3 | 83.9 | 86.4 | 51.8 | 25.5 | 65.4 | 57.8 |
| ODSW Easy only | 64.2 | 51.8 | 72.4 | 28.6 | 48.2 | 83.4 | 87.3 | 51.3 | 28.2 | 64.1 | 58.0 |
| ODSW Medium only | 68.2 | 51.9 | 74.7 | 29.9 | 44.2 | 83.9 | 88.3 | 52.4 | 27.9 | 64.6 | 58.6 |
| ODSW Hard only | 68.2 | 52.3 | 74.5 | 29.5 | 46.2 | 84.0 | 88.0 | 52.7 | 27.0 | 64.6 | 58.7 |
| VL-COGITO | 69.4 | 55.4 | 74.8 | 30.9 | 49.8 | 83.9 | 88.2 | 52.6 | 28.9 | 67.0 | 60.1 |

ScienceQA, MMStar), demonstrating its advantage when deeper reasoning is required. Adding progressive curriculum alone (*i.e.*, "+Curriculum") yields consistent gains across nearly all tasks, validating the general benefit of an easy-to-hard schedule. Incorporating dynamic length reward at the final stage further boosts results on the hard mathematical datasets (*e.g.*, MathVerse, LogicVista), raising the overall average to $60.1\%$. In contrast, applying DyLR directly to GRPO (*i.e.*, "+DyLR") destabilizes training, as the model prematurely interprets ill-formed queries as "difficult" and overextends its reasoning. This misalignment leads to inefficient learning, disrupted optimization, and weaker convergence and generalization.

**Impact of online difficulty soft weighting.** To assess how difficulty weighting methods affect VL-COGITO, we conduct an ablation using single-stage RL training with different weighting strategies (Table 3). *ODSW consistently outperforms binary weighting*, with larger gains under skewed difficulty distributions, likely because it maintains balanced optimization by adjusting emphasis across difficulty levels, whereas over-focusing on exclusively easy or hard samples impairs reasoning performance. Moreover, *adequate exposure to challenging queries is crucial*: models trained solely on easy cases underperform substantially, whereas softly prioritizing harder problems yields consistent gains. These findings inform our curriculum design: we adopt ODSW throughout and progressively increase the weight of challenging tasks in later stages to maximize reasoning proficiency.

**Impact of dynamic length reward.** Figure 2 visualizes the effect of length-reward strategies,

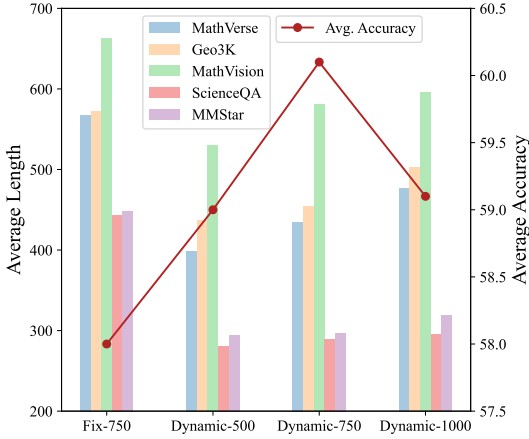

Figure 2: Performance of different length rewards. *Dynamic-N* applies a dynamic reward targeting length $N$ in the final stage of curriculum RL, whereas *Fix-N* enforces a fixed target length $N$ for all responses.

where we compare the dynamic length reward (Dynamic-$N$, $N \in \{500, 750, 1000\}$) with the fixed-length reward (Fix-$N$, $N = 750$) of Yeo et al. (2025), reporting average accuracy and response token counts on representative multimodal benchmarks. Unlike the fixed scheme, which uniformly pushes all outputs toward a preset length regardless of problem complexity, *dynamic reward adapts to task difficulty, yielding shorter and more concise reasoning on easy samples and selectively lengthen-*

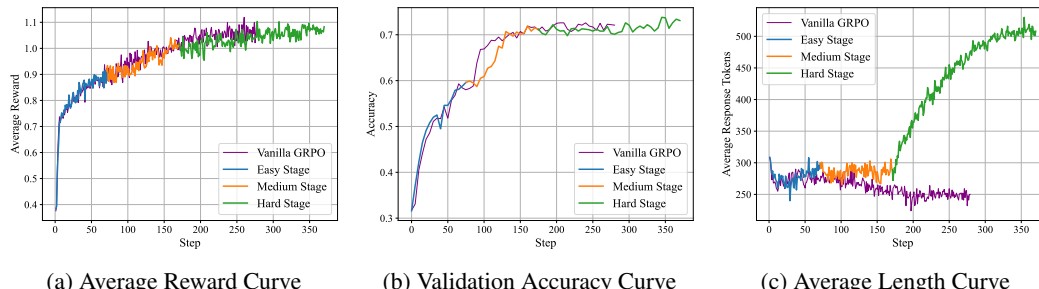

(a) Average Reward Curve      (b) Validation Accuracy Curve      (c) Average Length Curve

Figure 3: Training curves of PCuRL (target length 500 tokens) and vanilla GRPO, where (a) is average reward over sampled responses, (b) is validation accuracy on a held-out set ($\sim$1k questions) measured by the accuracy-based reward, and (c) denotes average response length.

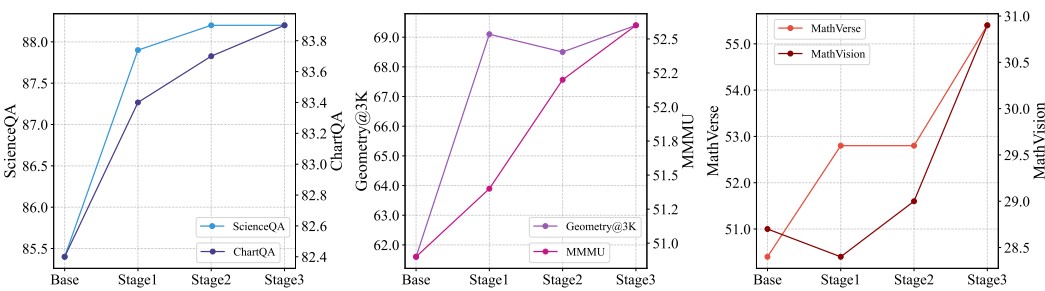

Figure 4: The performance of VL-COGITO across various training stages, evaluated on six benchmarks: ScienceQA, ChartQA, Geometry3K, MMMU, MathVerse, MathVision. These six benchmarks are arranged from easy to difficult, going from left to right.

*ing responses on hard ones*. As a result, Dynamic-$N$ consistently delivers higher average accuracy and a more context-appropriate length distribution, whereas Fix-$N$ often inflates responses with unnecessary steps without commensurate gains. Moreover, increasing the target length in Dynamic-$N$ ($500 \rightarrow 1000$) primarily extends outputs on math-heavy datasets—where deeper reasoning is beneficial—while lighter-reasoning benchmarks (*e.g.*, ScienceQA, MMStar) remain compact, confirming that dynamic rewards allocate length where it most improves performance.

**Training dynamics comparison between PCuRL and GRPO.** Figure 3a shows that PCuRL initially tracks the vanilla GRPO, indicating both optimize similar correctness-oriented rewards. Rewards rise steadily in the easy and medium stages. At the start of the hard stage, PCuRL briefly dips below the baseline because the dynamic length term penalizes outputs shorter than the increasing target; as the model adapts to produce longer responses, its reward rebounds to match the vanilla model. Validation accuracy (Fig. 3b) shows PCuRL matches GRPO in the easy and medium stages, rising from $0.3$ to $\sim 0.7$. In the hard stage, where responses lengthen (Fig. 3c), PCuRL surpasses the baseline—GRPO saturates near $0.7$ while PCuRL continues to a new peak—indicating that the longer reasoning chains materially improve problem solving rather than merely increasing output length. Figure 3c displays the average response length of both methods. Vanilla GRPO remains nearly flat at 250–300 tokens with a slight decline, indicating limited capacity to elicit longer or more diverse reasoning. By contrast, PCuRL adapts length to its curriculum: in the easy and medium phases it tracks the baseline, prioritizing correctness, while in the hard phase the dynamic length reward steadily increases the output from $\sim 280$ toward the 500-token target, demonstrating progressive guidance toward more elaborate responses. These results show that PCuRL's progressive curriculum effectively trains MLLMs to produce longer, more complex reasoning while improving validation accuracy. Unlike standard GRPO, which ignores response length, PCuRL's hard-stage dynamic length reward reliably reaches the target length and yields measurable accuracy gains.

**Performance of different training stages of PCuRL.** Figure 4 illustrates PCuRL's performance improvement from the base model to the final training stage. We display the performance against six benchmarks of varying difficulty, from easier ones on the left side to harder ones on the right side.

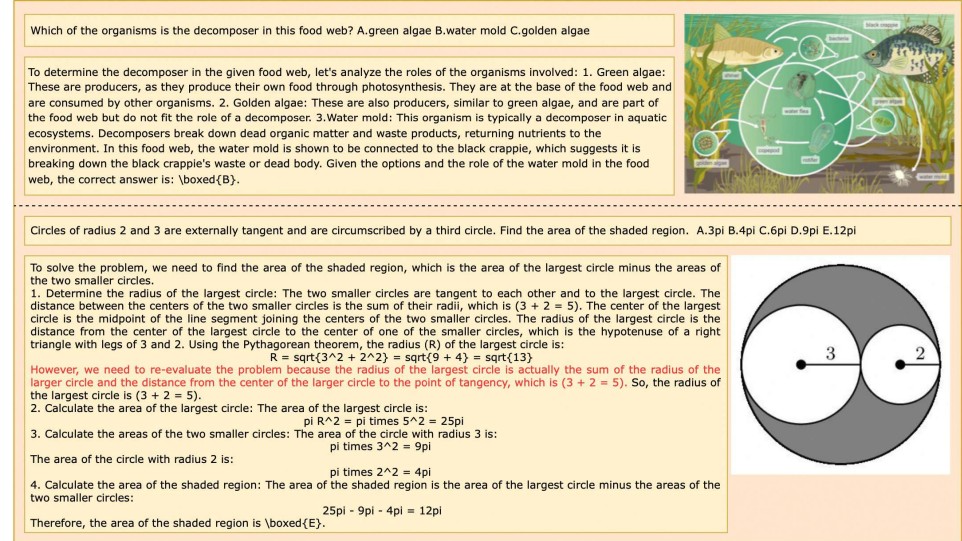

Figure 5: Case studies of VL-COGITO, where samples are drawn from ScienceQA and MathVision.

For a single benchmark, when the curve is skewed upward, it indicates that the model's performance improves rapidly in the early stages of training. Conversely, when the curve is skewed downwards, it suggests that the model achieves improvement mostly in the later stages. We can observe that as the difficulty of the benchmark increases, the performance curve becomes increasingly skewed downward. This suggests that more challenging problems often require more difficult stages to obtain improvements. This phenomenon aligns with the easy-to-hard training strategy of PCuRL, indicating the unique role of the different training stages in our curriculum learning framework.

## 5.3 CASE STUDY

Figure 5 illustrates multimodal reasoning cases of VL-COGITO. For multiple-choice questions in scientific domains, VL-COGITO systematically evaluates each option to identify the correct answer. In contrast, when addressing mathematical problems, the model typically first derives the solution independently and subsequently maps it to the given options. Notably, the model also exhibits self-reflection capabilities, as evidenced in the second example (highlighted in red in the figure). Initially, the model incorrectly applied the Pythagorean theorem when calculating the radius of the largest circle, leading to an erroneous result. However, it promptly recognized and addressed this error by invoking the "re-evaluate" mechanism, subsequently correcting its calculation and ensuring the accuracy of the remaining steps. This behavior underscores the effectiveness of our RL-based training pipeline in instilling valuable self-reflective abilities within multimodal models.

## 6 CONCLUSION

In this work, we introduce VL-COGITO, an advanced multimodal large language model (MLLM) enhanced by a progressive curriculum-based reinforcement learning framework termed PCuRL, designed to systematically improve multimodal reasoning capabilities. The multi-stage curriculum embedded within PCuRL progressively guides the model on tasks of increasing complexity, enhancing its proficiency in addressing reasoning challenges across various domains. A key component of our framework is an online difficulty soft weighting mechanism, which dynamically adjusts task difficulty in response to the model's evolving abilities, facilitating a balanced transition from simpler to more complex tasks. Furthermore, the dynamic length reward mechanism modulates the length of the model's responses according to problem-specific demands, optimizing the balance between reasoning depth and efficiency. Experimental results demonstrate that VL-COGITO achieves the state-of-the-art or highly competitive performance across various benchmarks, underscoring the substantial potential of meticulously crafted curriculum learning strategies to broaden the applicability of multimodal reasoning models.

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

## A    LIMITATIONS

While PCuRL demonstrates substantial gains in multimodal reasoning, several avenues remain open for further enhancement. Our current study is built upon Qwen2.5-VL-7B, and extending the framework to larger-scale MLLMs such as Qwen2.5-VL-32B would offer valuable validation of its scalability. Moreover, our work emphasizes a pure RL formulation, leaving the exploration of the conventional SFT-then-RL pipeline and its potential synergies with PCuRL as an interesting future direction. In addition, the stage-wise training schedule in PCuRL is currently determined empirically, whereas adaptive strategies that dynamically adjust stage transitions across models and datasets may further improve effectiveness. Such explorations hold promise for broadening the applicability of PCuRL to even more diverse scenarios.

## B    MORE TECHNICAL DETAILS

### B.1    DATA CURATION

To promote robust performance and strong generalization across a variety of domains, we curated an extensive collection of open-source multimodal datasets, covering 23 datasets distributed across six distinct task categories. (1) **Mathematical Reasoning**: Geometry@3K training set (hiyouga, 2025a), GeoQA+ (Cao & Xiao, 2022), Geos (Seo et al., 2015), GeomVerse (Kazemi et al., 2024), Inter-GPS (Lu et al., 2021), MultiMath (Peng et al., 2024); (2) **Logical Reasoning**: Raven (nimapourjafar, 2024), MM-IQ (Cai et al., 2025), EasyArc (Unsal & Akkus, 2025); (3) **Counting**: CLEVR-Math (Lindström & Abraham, 2022), Super-CLEVR (Li et al., 2023); (4) **Science Reasoning**: AI2D training set (Kembhavi et al., 2016), ScienceQA training set (Lu et al., 2022a), TQA (Kembhavi et al., 2017)); (5) **Chart Understanding**: ChartQA training set (Masry et al., 2022), TabMWP (Lu et al., 2023), DVQA (Kafle et al., 2018), FigureQA (Kahou et al., 2018), ArXivQA (Li et al., 2024b), InfographicVQA (Mathew et al., 2022); and (6) **General Image Understanding**: OKVQA (Marino et al., 2019), VQA2.0 (Antol et al., 2015), LLaVA-CoT (Xu et al., 2025).

For the data used in reinforcement learning, we selectively sample a subset of the data based on our designed quality and category criteria to construct the training set. During data curation, our primary objective is to enhance the overall difficulty and coverage of the training samples, encouraging the model to perform more in-depth reasoning. To this end, two additional measures are implemented: (1) **Open-ended Format**: To prevent the reasoning model from relying on superficial cues present in specific answer formats, such as multiple-choice, we reformulate most samples into an open-ended QA format; (2) **Difficulty Sampling**: To exclude questions that do not necessitate genuine reasoning, we employ difficulty-based sampling approach by removing samples that achieve above 50% accuracy over 8 trials using Qwen2.5-VL-7B-Instruct (Bai et al., 2025). The resultant distribution of RL training data following these filtering procedures is presented in Table 4.

### B.2    ONLINE DIFFICULTY SOFT WEIGHTING

Figure 6 illustrates the distributions of easy, medium, and hard stages of the online difficulty soft weighting (ODSW).

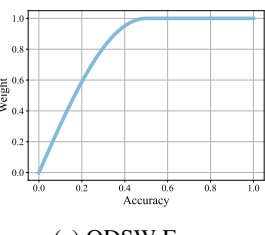 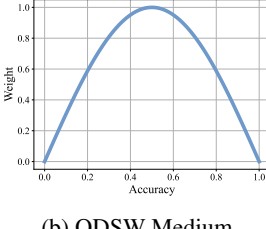 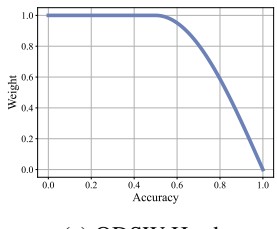

(a) ODSW Easy          (b) ODSW Medium          (c) ODSW Hard

Figure 6: Three difficulty distributions, *i.e.*, easy, medium, and hard, for the Online Difficulty Soft Weighting (ODSW).

Table 4: Data statistics of the training dataset in the RL stage, where the filter rate refers to the percentage of questions removed in the difficulty sampling, and the data size represents the number of samples after sampling.

| Category | QA Type | Dataset | Data size | Filter rate |
|---|---|---|---|---|
| Mathematics | Open-ended | Geometry3K | 2101 | 39% |
| | Open-ended | GeoQA+ | 7886 | 50% |
| | Open-ended | Geos | 367 | 22% |
| | Open-ended | GeomVerse | 8179 | 12% |
| | Open-ended | Inter-GPS | 946 | 26% |
| | Open-ended | MultiMath | 13503 | 35% |
| Logical | Open-ended | Raven | 6919 | 65% |
| | Open-ended | MM-IQ | 2492 | 7% |
| | Open-ended | EasyArc | 600 | 0% |
| Counting | Open-ended | CLEVR-Math | 1000 | 92% |
| | Open-ended | Super-CLEVR | 3000 | 20% |
| Science | Open-ended | AI2D | 5034 | 35% |
| | Multi-choice | ScienceQA | 1098 | 82% |
| | Open-ended | TQA | 5959 | 31% |
| Charts | Open-ended | ChartQA | 5570 | 72% |
| | Open-ended | TabMWP | 2963 | 70% |
| | Open-ended | DVQA | 2589 | 45% |
| | Open-ended | FigureQA | 2000 | 40% |
| | Open-ended | ArXivQA | 2000 | 57% |
| | Open-ended | InfographicVQA | 626 | 70% |
| General | Open-ended | OKVQA | 1500 | 12% |
| | Open-ended | VQA2.0 | 1500 | 22% |
| | Open-ended | LLaVA-CoT | 1500 | 8% |

## B.3 SYSTEM PROMPT

**Training Prompt:** We use this system prompt in RL training to encourage the model to follow the reasoning format and put the final answer in a "\boxed{}".

Table 5: The system prompt used for RL training.

A conversation between User and Assistant.

The User provides an image and asks a question. The Assistant first analyzes both the image and the question, then carefully thinks about the reasoning process step by step, and finally provides the User with an accurate answer. The Assistant must carefully checkout the correctness and validity of each reasoning step. If any errors or inconsistencies are found during the reasoning process, the Assistant reflects and corrects them logically.

The reasoning process and answer are enclosed within $<$ think $><$ /think $>$ and $<$ answer $><$ /answer $>$ tags, respectively, i.e., $<$ think $>$ detailed reasoning process here, with potential reflections and corrections $<$ /think $><$ answer $>$ final answer here, with the key result enclosed within \boxed{} $<$ /answer $>$

**Evaluation Prompt:** To ensure fairness during the evaluation, we adopt a simple prompt that can be generalized to most models rather than the system prompt used in the training.

Table 6: The system prompt used for evaluation.

> Please solve the problem step by step and put your answer in one "\boxed{}". If it is a multiple-choice question, only one letter is allowed in the "\boxed{}".

Table 7: Comparison of VL-COGITO with other MLLMs on multimodal reasoning benchmarks, including the scores (in parentheses) that are reported in the original papers. Results affected by benchmark contamination are excluded and marked as "–". The **bold** and underline highlight the best and second-best scores, respectively.

| Model | Size | Mathematics | | | | | | Science | | | General | Average |
|---|---|---|---|---|---|---|---|---|---|---|---|---|
| | | Geo3K | MathVerse | MathVista | MathVision | LogicVista | ChartQA | SciQA | MMMU | EMMA | MMStar | |
| *General-Purpose Models* | | | | | | | | | | | | |
| Qwen2.5-VL | 7B | 61.6 | 50.4 (49.2) | 69.3 (68.2) | 28.7 (25.1) | 44.0 | 82.4 | 85.4 | 50.9 | 24.6 | 62.5 (63.9) | 56.0 |
| InternVL2.5 | 8B | 60.6 | 40.0 (39.5) | 61.4 (64.4) | 19.9 (19.7) | 37.7 (36.0) | 73.4 | 90.3 | 43.1 (48.9) | 19.9 | 62.2 (62.8) | 50.9 |
| InternVL3 | 8B | 63.3 | 49.4 (39.8) | 68.5 (71.6) | 30.0 (29.3) | 41.3 (44.1) | 81.3 | 89.3 | 50.8 | 14.5 | 67.5 (68.2) | 55.6 |
| LLaVA-OV | 7B | 48.5 | 33.6 (26.2) | 56.4 (63.2) | 15.9 | 30.6 | 65.0 | 80.5 | 41.6 | 18.3 | 53.5 (61.7) | 44.4 |
| *Reasoning-Oriented Models* | | | | | | | | | | | | |
| MM-Eureka | 8B | 67.2 | 52.3 (50.3) | 73.4 (73.0) | 29.4 (26.9) | 47.1 | 82.7 | 86.4 | 52.3 | 27.4 | 64.7 | 58.3 |
| R1-VL | 7B | 57.5 | 41.3 (40.0) | 61.5 (63.5) | 23.0 (24.7) | 36.3 | 76.3 | 86.0 | 38.1 | 24.0 | 55.6 (60.0) | 50.0 |
| MMR1 | 7B | 65.9 | 52.5 (45.1) | 73.6 (71.0) | **32.9** (30.2) | 46.6 (50.8) | 82.8 | 86.7 | **53.1** | 28.1 | 66.3 | 58.9 |
| R1-OneVision | 7B | 57.9 | 44.0 (46.4) | 60.3 (64.1) | 22.0 (29.9) | 40.0 | 72.5 | 85.3 | 43.4 | 22.2 | 56.2 | 50.4 |
| OpenVLThinker | 7B | 60.6 | 48.1 (47.9) | 70.6 (70.2) | 22.0 (25.3) | 41.0 | 81.0 | 85.9 | 50.9 | 24.9 | 62.8 | 54.8 |
| VL-Rethinker | 7B | 67.7 | 54.6 (54.2) | 73.7 (74.9) | 30.1 (32.3) | 45.7 | 83.5 | 86.7 | 52.9 | 28.6 (29.7) | 64.2 | 58.8 |
| Vision-R1 | 7B | 67.0 | 51.9 (52.4) | 72.1 (73.5) | - | 44.7 | 82.7 | - | 26.4 | 28.3 | 65.4 | - |
| ThinkLite-VL | 7B | 63.5 | 51.3 (50.7) | 72.5 (75.1) | 27.5 | 44.3 | 83.1 | - | 50.9 | 26.4 | 64.6 (65.0) | - |
| VL-COGITO | 7B | **69.4** | **55.4** | **74.8** | 30.9 | **49.8** | **83.9** | **88.2** | 52.6 | **28.9** | **67.0** | **60.1** |

## C  MORE RESULTS

Table 7 summarizes the results that compare VL-COGITO with both general and reasoning MLLMs across 10 multimodal benchmarks, which are identical to Table 1. Nevertheless, Table 7 also includes the scores (in parentheses) that are reported in the original papers.

## D  LLM USAGE STATEMENT

In compliance with the ICLR 2026 policy on large language model (LLM) usage, we acknowledge that an LLM (OpenAI's ChatGPT) was employed in a **strictly limited, non-substantive capacity**. Its role was confined to *minor editorial support* (*e.g.*, grammar refinement and wording adjustments), suggestions regarding figure color schemes, and basic LaTeX formatting assistance. **The LLM did not contribute to the conception of research ideas, experimental design, data analysis, or substantive scientific writing**. All intellectual content and scientific contributions are solely attributable to the authors.

