# OpenReview forum: "Incentivizing Multimodal Reasoning via Progressive Curriculum Reinforcement Learning"
_ICLR.cc/2026/Conference — Submitted to ICLR 2026_

### Official Review · Reviewer_NkZZ · 2025-10-26

**Soundness:** 2
**Presentation:** 2
**Contribution:** 1
**Rating:** 2
**Confidence:** 4

**Summary:**

This paper proposes **VL-COGITO**, a MLLM designed for multimodal reasoning, trained via a **Progressive Curriculum Reinforcement Learning framework (PCuRL)**. PCuRL comprises two core components:

1. **ODSW**: Dynamically adjusts sample difficulty weights across different training stages, progressively enhancing the model’s reasoning capability from easy to hard examples.
2. **DyLR**: Adaptively modulates the length of reasoning chains based on task complexity, preventing over-reasoning on simple questions and under-reasoning on complex ones.

**Strengths:**

1. Introduces a curriculum-based reinforcement learning approach to equip the model with the ability to handle tasks of varying difficulty levels.

**Weaknesses:**

1. **Limited performance gains**: Although the method achieves state-of-the-art results on most benchmarks, the absolute improvement over the strongest baselines (e.g., VL-Rethinker, MMR1) is typically only 1–3 percentage points. On certain tasks, it even performs slightly worse, casting doubt on the statistical or practical significance of the reported gains.
2. **Scalability concerns**: PCuRL has only been validated on a 7B-parameter model. It remains unclear whether this framework generalizes effectively to much larger models (e.g., 30B+ parameters).

**Questions:**

1. Since the same dataset is used across all three training stages, could this lead to overfitting during the "hard" stage? Are there validation-set evaluations or cross-dataset generalization experiments to mitigate this concern?
2. Given the relatively modest performance improvements (mostly <3%), is the added complexity of integrating ODSW + DyLR with a multi-stage training pipeline justified?

---

> ### Author Response · Authors · 2025-11-18
> **Question Response**
>
> Thank you for your reviews and we would like to respond to the questions one by one.
>
> P1: Overfit problem on hard stage
>
> We would like to clarify that the total training process including all three stages approximately equal to only three epochs. Hence, the training iteration is far from overfit. At the same time, for each stage, we change the order of the data sequence to avoid overfitting in the data sequence. We also display the validation accuracy in the training in figure 3 (b), which shows the model converges in the hard stage.
>
> P2: Limited performance gains
>
> Considering we use ten benchmarks from various domains, we believe our overall improvement (average accuracy 4.1% higher than base model Qwen2.5-VL-7B) is significant. Comparing with Vanilla GRPO, the average performance gain is 2%, where the improvement mostly focus on harder math problems (3.2% on MathVerse and 3.4% on MathVista).
>
> As for the other most competitive comparison models, we outperform MMR1 1.2% and VL-Rethinker 1.3%. Moreover, for MMR1, it is not fair to directly compare our model with it. MMR1 add an additional SFT stage with cot data distilled from Qwen2.5-VL-72B, while our model is a completely RL approach. Even so, our model surpasses it in overall performance. For VL-Rethinker, it is a pure RL model, but its contribution is mainly focused on forced rethinking strategy in rollout, which does not conflict with our training framework.

---

### Official Review · Reviewer_6r76 · 2025-10-27

**Soundness:** 3
**Presentation:** 3
**Contribution:** 2
**Rating:** 2
**Confidence:** 4

**Summary:**

The paper proposes PCuRL (Progressive Curriculum Reinforcement Learning) and the model VL-COGITO. PCuRL organizes GRPO training into three stages—easy → medium → hard—over the same data, reshuffled per stage. There are two key mechanisms: (1) Online Difficulty Soft Weighting (ODSW): A stage-specific, continuous weighting of prompts by online rollout accuracy, (2) Dynamic Length Reward (DyLR): A cosine-shaped reward that targets, per prompt, the average length of correct rollouts (or a capped maximum when none are correct).

**Strengths:**

1. The paper is generally well-written and easy to follow, with a clear description of the method.
2. The paper provides intuitive visual demonstrations to help better understand the paper.

**Weaknesses:**

1. **Limited novelty.** The paper’s two core contributions: difficulty-aware weighting and a length reward within a curriculum RL framework have already been extensively explored in prior work [1,2,3]. Moreover, even the data curation choice of an open-ended response format has precedent (e.g., NoisyRollout) [4]. As a result, the manuscript appears to build primarily on established conclusions, and its incremental contributions seem limited relative to the existing literature.
2. **Design choice for online difficulty soft weighting.** In each curriculum stage, the authors reuse the same dataset and intervene via reweighting rather than performing the standard curriculum step of pre-partitioning samples by difficulty and staging them accordingly. Why not adopt the conventional approach of separating easy/medium/hard subsets and advancing through them? In addition, letting every stage train on the full data makes it difficult to attribute gains solely to the proposed soft weighting; improvements may stem from repeated exposure to all examples rather than from the weighting mechanism per se.
3. **Scope and evidence for the length reward.** The length reward is only activated in the hard stage. According to `Fig.3` (Training curves of PCuRL), the only clearly diverging trajectory from the baseline appears in the Average Length Curve during stage three. This design does not convincingly substantiate the claim in `Line 240–242` that the method “avoids fixed targets that risk over- or under-thinking.” A more interpretable analysis (*e.g.*, per-task length/accuracy Pareto fronts, error taxonomy vs. chain length) and a complete ablation across stages (length reward on/off in easy/medium/hard) are needed to validate this claim.
4. **Potential zero-gradient issue without filtering**. At `Line 172–173`, the authors emphasize that their approach “retains more prompts” instead of filtering as in other works. Does this choice reintroduce the all-correct / all-incorrect (i.e., zero-variance) groups that can yield zero or near-zero gradient under GRPO-style normalization?
5. **Fairness of experimental comparisons.** Fairness remains unclear in `Tab.1`. It is not specified whether baselines share the **same base model**; for instance, early versions of R1-VL [5] are based on **Qwen2-VL**, which differs from Qwen2.5-VL used here. Additionally, the paper samples **16** responses per prompt during training, whereas many compared methods use **8 or 12**. This hyperparameter materially affects group advantage estimation stability and accuracy in GRPO. Please report an ablation on the group size (G) and ensure base model parity to enable a fair comparison.
6. **Attribution to algorithm vs. data scale/overlap.** The training corpus spans many domains, with several sources closely related to the evaluation benchmarks (e.g., MathVista). By contrast, MM-Eureka [6] and NoisyRollout [4] train with substantially **smaller datasets** (≈15K and ≈3K, respectively). It remains unclear how much of the observed gains arise from the PCuRL algorithm versus data scale and domain overlap. Please provide experiments controlling for training data size and composition (e.g., fixed subsets, down-sampling to match [6]/[4]) to disentangle algorithmic benefits from data effects.

**References**

[1] GRPO-LEAD: A Difficulty-Aware Reinforcement Learning Approach for Concise Mathematical Reasoning in Language Models, https://arxiv.org/abs/2504.09696

[2] L1: Controlling How Long A Reasoning Model Thinks With Reinforcement Learning, https://www.arxiv.org/abs/2503.04697

[3] Boosting the Generalization and Reasoning of Vision Language Models with Curriculum Reinforcement Learning, https://arxiv.org/abs/2503.07065

[4] NoisyRollout: Reinforcing Visual Reasoning with Data Augmentation, https://arxiv.org/abs/2504.13055

[5] R1-VL: Learning to Reason with Multimodal Large Language Models via  Step-wise Group Relative Policy Optimization, https://arxiv.org/abs/2503.12937v1

[6] MM-Eureka: Exploring the Frontiers of Multimodal Reasoning with Rule-based Reinforcement Learning, https://arxiv.org/abs/2503.07365

**Questions:**

See the `Weaknesses` part.

---

> ### Author Response · Authors · 2025-11-18
> **Question Response**
>
> Thank you for your reviews and we would like to respond to the questions one by one.
>
> P1: Novelty
>
> Our framework aims to conduct reasoning on multimodal tasks across different domains and difficulty levels. Although there is difficulty-weight/filtering approach and curriculum learning framework in RL, we are the first to combine the two component to form a online weighting curriculum learning framework. And this is crucial for our initial target (we discuss why online curriculum learning is better than conventional curriculum learning in problem 2). For DyLR, most previous length reward serve for a single purpose, either extend reasoning length or compress it, while our approach propose a novel reward that encourge model to find appropriate reasoning length. This allows the model to continue exploring difficult problems while maintaining efficiency on simple ones. It serves the main idea of our paper, learning reasoning on task with totally different difficulty levels.
>
> P2: Online difficulty soft weighting
>
> The limitation for conventional approach is that it can not avoid task bias caused by difficulty, which may cause catastrophic forgetting on certain tasks. Meanwhile, it also ignore the fact that the difficulty of the problem is constantly changing based on the model training process. Determine problem difficulty in every training step is more effective than separate them before the training. At last, we would like to clarify that the total training process including all three stages approximately equal to only three epochs. So it is hard to say that the improvement gain from repeated exposure to all examples.
>
> P3: Scope and evidence for the length reward
>
> For line 240-242, the supporting experimental evidences are in Figure 2, where we display the comparison between fix-length reward and our DyLR. We display the per-task length and overall accuracy, and we are willing to add per-task accuracy in the appendix. Unlike the fix-length reward, which uniformly pushes all outputs toward a preset length regardless of problem complexity, DyLR adapts to task difficulty, yielding shorter and more concise reasoning on easy samples and selectively lengthening responses on hard ones. Even when the target length increase, the response length for easier tasks still maintain around 300 tokens.
> For applying DyLR on easy stage and medium stage, we have done related experiment. We find that the average response length will not increase like the hard stage. The reason is that all-incorrect cases contribute a lot to exploring longer reasoning length, and the weight for all-incorrect cases are set to 0 in easy and medium stage. In this case, we use DyLR mainly in the hard stage.
>
> P4: Potential zero-gradient issue without filtering
>
> We would like to clarify that after we add length reward in hard stage, the gradient for all-correct/all-incorrect cases are not zero. More importantly, all-incorrect cases are particular crucial for exploration on longer reasoning length. In medium stage, the weight all-correct/all-incorrect cases are 0. In easy stage, the weight for all-incorrect cases are 0. In these situation, all-correct/all-incorrect cases only brings some minor decrease on training efficiency, which is not a big problem.
>
> P5: Fairness of experimental comparisons
>
> We would like to add an addition column in table 1 for base model. Notably, apart from the R1-VL(it contains two version), all the other models are based on qwen2.5-vl, which makes the comparison fair enough.  As for rollout group size, we list this hyperparameter from our comparison models (if provided): MM-Eureka 8, R1-VL 4, MMR1 32, VL-Rethinker 8, Vision-R1 16/8 for two stages. In this case, we don't think 16 is a outrageous hyperparameter.
>
> We conduct a new experiment about rollout group size on our framework, using rollout 8 as the reviewer suggested. It shows tha model with rollout 8 achieve a 59.7% average accuracy on all benchmarks, 0.4% lower that our original model 60.1%. The rollout group size will not have a huge impact on model performance. So it is not fair to say that the improvement of PCuRL is only rely on bigger rollout group size.
>
> P6: Attribution to algorithm vs. data scale/overlap
>
> To disentangle algorithmic benefits and data effect, we can compare the PCuRL+our data and Vanilla GRPO+our data in table 2. All of our comparison models have their own training set for RL, even with additional sft dataset like MMR1 (we do not have this). It is hard to maintain the absolute fairness on the data level.
>
> We conduct a new experiment on smaller data scale. Here, we randomly sample 20k data from our original data as a smaller dataset, containing 8.7k math data and 11.3k data from more generalized domains (MM-Eureka use 15k high quality math data). The model trained on this smaller dataset achieve a 59.4% average accuracy on all benchmarks, 0.7% lower that model with full dataset (60.1%). This model still outperforms models like MM-Eureka (58.3%).

---

### Official Review · Reviewer_faaJ · 2025-10-30

**Soundness:** 3
**Presentation:** 3
**Contribution:** 2
**Rating:** 6
**Confidence:** 3

**Summary:**

The paper proposes VL-COGITO, which uses a three-stage progressive curriculum to reinforce learning (easy → medium → difficult) to train a multimodal reasoning model: a dynamic length reward is introduced in the difficult stage, with the average thought length of each correct sample as the target, which encourages deeper thinking for complex questions and avoids verbose thinking for simple questions; it achieves 8 state-of-the-art results on 10 benchmarks and close results for the rest, showing that accuracy and stability can be significantly improved without SFT.

**Strengths:**

This paper employs a course-based reinforcement learning approach, progressing from easy to medium to difficult tasks. It first stabilizes and optimizes quickly on easier problems, then gradually focuses on more challenging tasks, significantly reducing training oscillations while improving gradient and data utilization.

In the difficult phase, a dynamic reasoning length reward is introduced, setting a target based on the average inference length of correct samples per problem. Complex problems encourage deeper thinking, while simple problems maintain conciseness.

Eight of the ten multimodal benchmarks achieved SOTA results, with the rest approaching or surpassing the benchmarks.

**Weaknesses:**

Are there any automated switching methods for multi-stage training, or what quantitative metrics can guide multi-stage training?

Does encouraging longer, more challenging problem-solving sessions increase the likelihood of issues like repeated output or random output? Should we add penalties for this?

Figure 3 shows that the improvement in the last stage is not significant, and the increase in the estimated length in the third stage has no obvious effect.

**Questions:**

Please see the weaknesses

---

> ### Author Response · Authors · 2025-11-18
> **Question Response**
>
> Thank you for your reviews and we would like to respond to the questions one by one.
>
> P1: Switching methods for multi-stage training
>
> The stage transition strategy consists of two main parts. First, we train the model in each stage until it reaches near-convergence (on validation accuracy and average response length). Second, we select the checkpoint with the best validation accuracy near the converged step as the starting point for the next one. And we also believe that the automated switching strategy is one of the future direction of this work, and we would like to conduct further investigation on it.
>
> P2: Possibility for repeated and random output
>
> During the training, we do not observe issue like repeated or random output. But we do find that the response become verbose if the average response length grow too fast in the training, and we have two mechanisms to suppress this. When the average response length growth is too fast, the model cannot learn to effectively utilize complex reasoning with extra length, but instead adapts excessively to lengthy reasoning patterns that are only used to increase response length. The growth rate of average response length is mainly determined by two things: (1) the proportion of cases with acc=0 (2) w in equation 8. We observe this verbose issue in the experiment of Vanilla GRPO+DyLR in table 2, since there exist a large proportion of cases with acc=0 in the begining of the training.
> To avoid this rapid growth of respnse length, we use DyRL only in stage 3 of the training (when cases with acc=0 decrease) and set w to a small number 0.25. As the average response length grows slowly, the model has sufficient steps to learn how to use increased length for effective reasoning. Hence, we find external penalties not necessary in our framework.
>
> P3: Clarifying the gain of the last stage
>
> The improvement of the stage 3 mainly focuses on the hard questions, which is shown in Figure 4. Figure 4 display a comparison between the performance of the three different training stages on easy to hard test set. We can observe that in hard math test set like mathvision and mathverse (on the right), the stage 3 achieve a major improvement (1.9 on mathvision and 2.6 on mathverse). There are two reasons why the improvement in the stage 3 is not significant in Figure 3. (1) The wide scale of y-axis makes the improvement (from 72 to 74) not as obvious. (2) Figure 3 is based on the validation set, a split from the training set, where easier questions occupy a relatively large proportion.

---

> > ### Comment · Reviewer_faaJ · 2025-11-28
> >
> > The author's reply resolved my question, so I chose to increase my score.

---

### Official Review · Reviewer_24BX · 2025-11-01

**Soundness:** 3
**Presentation:** 3
**Contribution:** 3
**Rating:** 6
**Confidence:** 4

**Summary:**

The paper proposes a multimodal reasoning model named VL-COGITO, whose core is a novel PCuRL framework. This framework aims to address the performance issues arising from the complexity and diversity of multimodal tasks. PCuRL features two key innovations: 1. an ODSW mechanism that dynamically adjusts task difficulty during training. 2. a DyLR mechanism that incentivizes the model to adaptively control the length of its reasoning path according to the complexity of each problem.
Experimental results show that VL-COGITO achieves state-of-the-art performance on 8 out of 10 benchmark tasks, and the authors further conduct extensive ablation studies to verify the effectiveness of each component.

**Strengths:**

1. This paper combines curriculum learning with reinforcement learning to specifically tackle complex multimodal reasoning problems. The two core components ODSW and DyLR are designed to address challenges in training stability and reasoning depth across tasks of varying difficulty. This approach provides valuable guidance for training multimodal reasoning models.
2. The paper conducts extensive experiments on diverse multimodal benchmarks spanning mathematics, science, logic, and general understanding. VL-COGITO achieves SOTA performance on eight benchmarks, outperforming both general models and other SOTA reasoning models. The authors also perform comprehensive ablation studies to demonstrate the effectiveness of the proposed methods.

**Weaknesses:**

1. The authors claim in line 60 that the model “bypasses the cold-start SFT phase.” However, the backbone is Qwen2.5-VL-7B-Instruct, which has already undergone extensive SFT on large-scale instruction-following data. Thus, this statement is misleading, as the model benefits from a SFT procedure.
2. During data curation, the authors discard samples with pass@8 > 50%, keeping only data with higher "learnability". I am curious how a simple SFT on these selected  samples would perform compared to the proposed PCuRL pipeline.
3. The paper states (around line 350) that applying DyLR directly to GRPO destabilizes training. However, in Table 2, the model “+DyLR” actually outperforms vanilla GRPO (58.1 → 58.6). This discrepancy is confusing and needs clarification.
4. DyLR (Eq. 5 & Eq. 8)
   When Acc = 0, the DyLR encourages the model to generate longer responses via L_max, while Equation 8 simultaneously down-weights these samples with w = 0.25. This appears contradictory: on one hand incentivizing exploration, on the other reducing its significance.

   I would like to know:

   a. Has the authors verified whether this design truly produces conflicting optimization signals?

   b. Since DyLR is only applied in the final stage, where accuracy should already be higher (if not, please clarify), are these potentially conflicting cases (i.e., Acc = 0) too rare to matter, and thus masked by the accuracy gains from earlier stages?

5. The difficulty-dependent weighting functions used in ODSW appear heuristic. The paper lacks an in-depth analysis for these functional forms.

**Questions:**

See weaknesses

---

> ### Author Response · Authors · 2025-11-18
> **Question Response**
>
> Thank you for your reviews and we would like to respond to the questions one by one.
>
> P1: Clarifying "Bypassing the cold-start SFT phase"
>
> Thanks for pointing this out and we will adjust the wordings. Some of our comparison model (MMR1, R1-OneVision, OpenVLThinker, Vision-R1) go through an extra long chain-of-thought (COT) SFT phase before RL, and the "SFT" in our context by default denotes this extra long COT SFT phase, which is aimed to inject reasoning patterns into the model.
>
> P2: SFT experiment
>
> As the reviewer suggested, we have done the pure SFT experiment from Qwen2.5-VL using our training data. The average accuracy on all benchmark is 51.3, which is even worse than Qwen2.5-VL itself (56). We believe the main reason is that we do not have any cot thinking data related to our training set, so we only use the groundtruth answer as training label. This makes the model overfit to a non-thinking mode that directly output the answer with zero thinking process, which is harmful for its performance on difficult math benchmarks.
>
> P3: Clarifying "applying DyLR directly to GRPO destabilizes training"
>
> Here, we want to highlight that DyLR offers a higher improvement over GRPO+Curriculum (58.9 to 60.1) compared to Vanilla GRPO (58.1 to 58.6). Meanwhile, applying DyLR directly to GRPO does bring some problems. Considering the large proportion of cases with acc=0 in the start of the training, it leads to a rapid growth in the average response length. In this case, the model tends to overfit the long reasoning pattern even for the easy questions (performance drops on easier datasets like ChartQA, ScienceQA, MMStar).
>
> P4:  L_max and w in DyLR
>
> Both L_max and w are crucial for exploration, but on different aspect. In fact, we have the experiment of how L_max and w influence the model's average response length, and we would like to add them in the appendix. The increase of L_max leads to a higher exploration upper bound (a higher average response length after converging) while maintaining a similar average response length growth rate. Meanwhile, a larger w suggests a higher growth rate of average response length. Notably, these experiments are done on the stage 3 of PCuRL. Here, w has a great impact on the training process (w increasing from 0.25 to 0.5 doubles the growth rate). This suggests that cases with acc=0 still occupies a considerable proportion of data in stage 3.
> The main reason that we want to control the growth rate of average response length is partly discussed in point 3 above. A steep gradient for response length results in generating verbose outputs, rather than utilizing the additional length for more sophisticated reasoning.
>
> P5:  Functional forms of difficulty-dependent weighting
>
> We thank the reviewer for this observation. The current weighting function was chosen for its simplicity and compatibility with our focus of "online", "continuous" weighting mechanism based on difficulty. While we acknowledge the heuristic nature of this choice, a thorough ablation or theoretical analysis of these forms is beyond our current scope, especially given the high computational cost of RL experiments. We leave it as an important direction for future research, much like how learning rate scheduling evolved from linear to more sophisticated strategies.

---

### Meta-Review · Area_Chair_eShm · 2026-01-08

**Summary:**

- The main concerns are summarized as follow:

1. Limited Novelty and Contribution:
The reviewer felt that the contributions of the paper were not sufficiently novel. The combination of difficulty-aware weighting and dynamic length rewards within a curriculum reinforcement learning framework has been explored in prior work.

2. Performance Gains and Statistical Significance:
While VL-COGITO achieved state-of-the-art results on most benchmarks, the performance improvements were seen as limited.

3. Scalability Concerns:
The reviewer highlighted that the proposed framework was only validated on a 7B-parameter model, and its generalizability to much larger models (e.g., 30B+ parameters) was unclear.

**Reviewer Concerns:**

- Concerns Still Outstanding:

1. Limited Novelty:
The reviewer still felt that the paper’s contributions were incremental, given the prior existence of similar approaches. The rebuttal did not fully resolve this concern, as the claims of novelty rely on the authors' framing of their specific combination of methods.

2. Scalability:
The scalability of the framework to larger models was not addressed in the rebuttal. Given that the method was only validated on a 7B-parameter model, this remains a critical concern for the applicability of the framework to larger, more complex multimodal models.

3. Performance Gains:
The reviewers are left questioning whether the added complexity of PCuRL justifies these modest gains. The rebuttal did not offer further experimental evidence that would make these small improvements more compelling.

**Reviewer Scores:**

None

---

### Decision · Program_Chairs · 2026-01-26

Reject